# POWERS OF LAYERS FOR IMAGE-TO-IMAGE TRANSLATION

## ABSTRACT

We propose a simple architecture to address unpaired image-to-image translation tasks: style or class transfer, denoising, deblurring, deblocking, etc. We start from an image autoencoder architecture with fixed weights. For each task we learn a residual block operating in the latent space, which is applied iteratively until the target domain is reached. A specific training schedule is required to alleviate the exponentiation effect of the iterations. At test time, it offers several advantages: the number of weight parameters is limited and the strength of the transformation can be modulated simply with the number of iterations. This is useful, for instance, when the type or amount of noise to suppress is not known in advance. Experimentally, we show that the performance of our model is comparable or better than CycleGAN and Nice-GAN with fewer parameters.

## 1 INTRODUCTION

Neural networks define arbitrarily complex functions involved in discriminative or generative tasks by stacking layers, as supported by the universal approximation theorem (Hornik et al., 1989; Montúfar, 2014). More precisely, the theorem states that stacking a number of basic blocks can approximate any function with arbitrary precision, provided it has enough hidden units, with mild conditions on the non-linear basic blocks. Studies on non-linear complex holomorphic functions involved in escape-time fractals showed that iterating simple non-linear functions can also construct arbitrarily complex landscapes (Barnsley et al., 1988). These functions are complex in the sense that their iso-surfaces are made arbitrarily large by increasing the number of iterations. Yet there is no control on the actual shape of the resulting function. This is why generative fractals remain mathematical curiosities or at best tools to construct intriguing landscapes.

In this paper, our objective is to combine the expressive power of both constructions, and we study the optimization of a function that iterates a single building block in the latent space of an auto-encoder. We focus on image translation tasks, that can be trained from either *paired* or *unpaired* data. In the paired case, pairs of corresponding input and output images are provided during training. It offers a direct supervision, so the best results are usually obtained with these methods (Chen et al., 2017; Wang et al., 2018; Park et al., 2019). We focus on **unpaired translation**: only two corpora of images are provided, one for the input domain $\mathcal{A}$ and the other for the output domain $\mathcal{B}$. Therefore we do not have access to any parallel data (Conneau et al., 2017), which is a realistic scenario in many applications, e.g., image restoration. We train a function $f_{\mathcal{AB}} : \mathcal{A} \to \mathcal{B}$, such that the output $b^* = F(a)$ for $a \in \mathcal{A}$ is indiscernible from images of $\mathcal{B}$.

Our transformation is performed by a single residual block that is composed a variable number of times. We obtain this compositional property thanks to a progressive learning scheme that ensures that the output is valid for a large range of iterations. As a result, we can modulate the strength of the transformation by varying the number of times the transformation is composed. This is of particular interest in image translation tasks such as denoising, where the noise level is unknown at training time, and style transfer, where the user may want to select the best rendering. This "Powers of layers" (PoL) mechanism is illustrated in Figure 1 in the category transfer context (horse to zebra).

Our architecture is very simple and only the weights of the residual block differ depending on the task, which makes it suitable to address a large number of tasks with a limited number of parameters. This proposal is in sharp contrast with the trend of current state-of-the-art works to specialize the architecture and to increase its complexity and number of parameters (Fu et al., 2019; Viazovetskyi et al., 2020; Choi et al., 2020). Despite its simplicity, our proof of concept exhibits similar or better performance than a vanilla CycleGAN architecture, all things being equal otherwise, for the

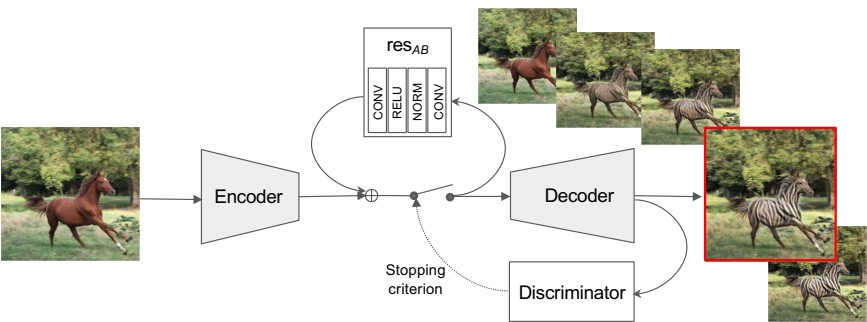

Figure 1: Illustration of Powers of layers for a category transfer task. The encoder and decoder are directly borrowed from a vanilla auto-encoder and are not learnable. At inference time, we apply a variable number of compositions, producing different images depending on how many times we compose the residual block in the embedding space. Depending on the task, we either modulate the transformation and choose the result, or let a discriminator determine when to stop iterating.

original set of image-to-image translation tasks proposed in their papers, as well as for denoising, deblurring and deblocking. With significantly fewer parameters and a versatile architecture, we report competitive results confirmed by objective and psycho-visual metrics, illustrated by visualizations. We will provide the implementation for the sake of reproducibility.

## 2 RELATED WORK

**Generative adversarial networks (GANs)** (Goodfellow et al., 2014) is a framework where two networks, a generator and a discriminator, are learned together in a zero-sum game fashion. The generator learns to produce more and more realistic images w.r.t. the training dataset. The discriminator learns to separate between real data and increasingly realistic generated images. GANs are used in many tasks such as domain adaptation (Bousmalis et al., 2016), style transfer (Karras et al., 2019b), inpainting (Pathak et al., 2016) and talking head generation (Zakharov et al., 2019).

**Unpaired image-to-image translation** considers the tasks of transforming an image from a domain $\mathcal{A}$ into an image in a domain $\mathcal{B}$. The training set comprises a sample of images from domains $\mathcal{A}$ and $\mathcal{B}$, but no pairs of corresponding images. A classical approach is to train two generators ($\mathcal{A} \rightarrow \mathcal{B}$ and $\mathcal{B} \rightarrow \mathcal{A}$) and two discriminators, one for each domain. When there is a shared latent space between the domains, a possible choice is to use a variational auto encoder like in CoGAN Liu & Tuzel (2016). CycleGAN Zhu et al. (2017), DualGAN Yi et al. (2017) and subsequent works (Liu et al., 2019; Fu et al., 2019; Choi et al., 2020) augment the adversarial loss induced by the discriminators with a cycle consistency constraint to preserve semantic information throughout the domain changes. All these variants have architectures roughly similar to CycleGAN: an encoder, a decoder and residual blocks operating on the latent space. They also incorporate elements of other networks such as StyleGAN Karras et al. (2019a). In our work, we build upon a simplified form of the CycleGAN architecture that generalizes over tasks easily. We applied also our method to NICE-GAN Chen et al. (2020). A concurrent work, GANHopper by Lira et al. (2020) proposes to iterate CycleGAN generators in order to perform the transformation. However, their method has many difference with ours: They iterate full generators and not a single residual block, their encoder and decoder are not fixed, their number of iteration is fixed and they have to use additional discriminators to act on intermediate transformation states. Other works (Zhang et al., 2019; Li et al., 2020) use recurrent networks to perform transformations but this is done in a paired context, which results in very different methods from ours and GANHopper.

**Transformation modulation** is an interpolation between two image domains. It is a byproduct of some approaches:For instance, a linear interpolation in latent space (Brock et al., 2018; Radford et al., 2015) morphs between two images. Nevertheless, one important limitation is that the starting and ending points must both be known, which is not the case in unpaired learning. Other approaches such as the Fader networks (Lample et al., 2017) or StyleGan2 (Viazovetskyi et al., 2020) act on scalar or boolean attributes that are disentangled in the latent space (*eg.*, age for face images, wear glasses or not, etc). Nevertheless, this results in complex models, for which dataset size and the variability of

images strongly impacts the performance: they fail to modulate the transform with small datasets or with large variabilities. A comparison of PoL with the Fader network is provided in Appendix C and shows that our approach is more effective.

**Progressive learning and inference time modulation** are performed in multi-scale methods such as SinGAN (Rott Shaham et al., 2019) and ProgressiveGAN (Karras et al., 2017). Progressive learning obtains excellent results for high resolution images where it is more difficult to use classical approaches. The training is performed in several steps during which the size of both the images and the network are increased. The inference time of some architectures can be modulated by stopping the forward pass at some layer (Huang et al., 2016; Wu et al., 2017; Veit & Belongie, 2017).This differs from our approach, where the number of residual block compositions ("powers") can be chosen, to shorten the inference. A by-product is a reduction of the number of network parameters.

**Weight sharing** is a way of looking at our method, because the same layer is applied several times within the same network. Recurrent Neural Networks (RNN) are the most classical example weight sharing in a recursive architecture. Besides RNNs, weight sharing is mainly used for model compression, sequential data and ordinary differential equations (ODE) (Gao et al., 2019; Han et al., 2015; Chen et al., 2018).A few works (Jastrzebski et al., 2017; Zhang & Jung, 2018) apply weight sharing to unfold a ResNet and evaluate its performance in classification tasks. The optimization is inherently difficult, so they use independent batch normalization for each shared layer. With PoL we observe the same optimization issues, that we solve by a progressive training strategy, see Section 3.3. Recent work (Jeon et al., 2020) are interested in the composition of the same block by considering the parallel with the fixed point theorem, nevertheless their application remains to rather simple problems compared to our unpaired image-to-image translation tasks.

## 3 POWER OF LAYERS

We adopt the same context as CycleGANand focus on unpaired image-to-image translation: the objective is to transform an image from domain $\mathcal{A}$ into an image from domain $\mathcal{B}$. In our case the domains can be noise levels, painting styles, blur, JPEG artifacts, or simply object classes that appear in the image. The training is unpaired: we do not have pairs of corresponding images at training time. CycleGAN is simple, adaptable to different tasks and allows a direct comparison in Sections 4 and 5.

We learn two generators and two discriminators. The generator $G_{AB} : I \to I$ transforms an element of $\mathcal{A}$ into an element of $\mathcal{B}$, and $G_{\mathcal{BA}}$ goes the other way round, $I$ being the fixed-resolution image space. The discriminators $D_{\mathcal{A}} : I \to [0,1]$ (resp. $D_{\mathcal{B}}$) predicts whether an element belongs to domain $\mathcal{A}$ (resp $\mathcal{B}$). We use the same losses as commonly used in unpaired image-to-image translation:

$$\mathcal{L}_{\text{Total}} = \lambda_{\text{Adv}}\mathcal{L}_{\text{Adv}} + \lambda_{\text{Cyc}}\mathcal{L}_{\text{Cyc}} + \lambda_{\text{Id}}\mathcal{L}_{\text{Id}}, \tag{1}$$
$$\text{where } \mathcal{L}_{\text{Adv}}(G_{AB}, D_B) = \mathbb{E}_{b\sim\mathcal{B}}[\log D_B(b)] + \mathbb{E}_{a\sim\mathcal{A}}[\log(1 - D_B(G_{AB}(a)))],$$
$$\mathcal{L}_{\text{Cyc}}(G_{AB}, G_{BA}) = \mathbb{E}_{b\sim\mathcal{B}}[\|G_{AB}(G_{BA}(b)) - b\|_1] + \mathbb{E}_{a\sim\mathcal{A}}[\|G_{BA}(G_{AB}(a)) - a\|_1],$$
$$\mathcal{L}_{\text{Id}}(G_{AB}, G_{BA}) = \mathbb{E}_{b\sim\mathcal{B}}[\|G_{AB}(b) - b\|_2] + \mathbb{E}_{a\sim\mathcal{A}}[\|G_{BA}(a) - a\|_2].$$

The Adversarial loss $\mathcal{L}_{\text{Adv}}(G_{AB}, D_B)$ verifies that the generated images are in the correct domain. The Cycle Consistency loss $\mathcal{L}_{\text{Cyc}}(G_{AB}, G_{BA})$ ensures a round-trip through the two generators reconstructs the initial image, and the identity loss $\mathcal{L}_{\text{Id}}(G_{AB}, G_{BA})$ penalizes the generators transforming images that are already in their target domain. We keep the same linear combination coefficients as in CycleGAN:$\lambda_{\text{Adv}} = 1$, $\lambda_{\text{Cyc}} = 10$, $\lambda_{\text{Id}} = 5$.

### 3.1 NETWORK ARCHITECTURE

We start from the CycleGAN architecture.The encoder and decoder consist of 2 layers and a residual block. The embedding space $\mathcal{E}$ of our model is $256 \times 64 \times 64$: its spatial resolution is 1/4 the input image resolution of $256 \times 256$ and it has 256 channels. All translation operations take place in the fixed embedding space $\mathcal{E}$. The encoder $\text{Enc} : I \to \mathcal{E}$ produces the embedding and consists of two convolutions. The decoder $\text{Dec} : \mathcal{E} \to I$ turns the embedding back to image space and consists of two transposed convolutions.

**Pre-training of the auto-encoder.** We train the encoder and decoder of our model on a reconstruction task using an $\ell_2$ reconstruction loss in pixel space. We use the Adam optimizer. Our

data-augmentation consists of an image resizing, a random crop and a random horizontal flip. Both the encoder and decoder weights are fixed for all the other tasks, only the residual block is adapted (and the discriminator in case we use it for the stopping criterion).

There are two choices to train the auto-encoder: (1) train with 6M unlabeled images randomly drawn from the YFCC100M dataset (Thomee et al., 2016) during one epoch, which makes it independent of the translation training dataset; (2) train on the dataset of our unpaired image-to-image translation task. For simplicity, we choose the first option and use a single encoder and decoder for all the tasks presented in this article. For the NiceGAN experiments, the encoder and decoder are trained jointly with the discriminator, so the auxiliary dataset YFCC100M is not used.

**The embedding transformer – single block.** The transformation between domains is based on a residual block $f_{\mathcal{AB}}$, similar to the feed-forward network used in transformers (Vaswani et al., 2017). It writes:

$$f_{\mathcal{AB}}(x) = x + \text{res}_{\mathcal{AB}}(x), \forall x \in \mathcal{A}. \tag{2}$$

There is a dimensionality expansion factor $K$ between the two convolutions in the residual block (see Figure 1). Adjusting $K$ changes the model's capacity. We adopt the now standard choice of the original transformer paper ($K = 4$). The full generator writes

$$G_{\mathcal{AB}}(x) = \text{Dec}(f_{\mathcal{AB}}(\text{Enc}(x))), \forall x \in \mathcal{A}. \tag{3}$$

The other direction, with $f_{\mathcal{BA}}$ and $\text{res}_{\mathcal{BA}}$, is defined accordingly.

**Powers of layers.** We start from the architecture above and augment its representation capacity. There are two standard ways of doing this: (1) augmenting the capacity of the $f_{\mathcal{AB}}$ block by increasing $K$; (2) increasing the depth of the network by chaining several instances of $f_{\mathcal{AB}}$, since the intermediate representations are compatible. In contrast to these fixed architectures, PoL *iterates* the $f_{\mathcal{AB}}$ block $n \geq 1$ times, which amounts to sharing the weights of a deeper network:

$$G_{\mathcal{AB}}(x) = \text{Dec}(f_{\mathcal{AB}}^n(\text{Enc}(x))), \forall x \in \mathcal{A}. \tag{4}$$

### 3.2 Optimization in a residuals blocks weight sharing context

In the following, we drop the $\mathcal{AB}$ suffix from $f_{\mathcal{AB}}$, since powers of layers operates in the same way on $f_{\mathcal{AB}}$ and $f_{\mathcal{BA}}$. Thus, $f : \mathcal{E} \to \mathcal{E}$ is $f(x) = x + \text{res}(x)$. The parameters of $f$ are collected in a vector $w$. The embedding $x \in \mathcal{E}$ is a 3D activation map, but for the sake of the mathematical derivation we linearize it to a vector. The partial derivatives of $f$ are $\frac{\partial f}{\partial x} = \frac{\partial \text{res}}{\partial x} + \text{Id}$ and $\frac{\partial f}{\partial w} = \frac{\partial \text{res}}{\partial w}$. We compose the $f$ function $n$ times as

$$\frac{\partial f^n}{\partial x}(x) = \prod_{i=n-1}^{0} \frac{\partial f}{\partial x}(f^i(x)) \quad \text{and} \quad \frac{\partial f^n}{\partial w}(x) = \prod_{i=n-1}^{1} \frac{\partial f}{\partial x}\left(f^i(x)\right)\frac{\partial f}{\partial w}(x). \tag{5}$$

The stability of the SGD optimization depends on the magnitude and conditioning of the matrix:

$$M_n = \prod_{i=n-1}^{1} \frac{\partial f}{\partial x}\left(f^i(x)\right) = \prod_{i=n-1}^{1}\left(\frac{\partial \text{res}}{\partial x}\left(f^i(x)\right) + \text{Id}\right), \tag{6}$$

which is sensitive to initialization during the first optimization epochs. Indeed, the length of the SGD steps on $w$ depends on the eigenvalues of $M_n$. When simplifying the basic residual block to a linear transformation $L \in \mathbb{R}^{d \times d}$ (i.e., ignoring the normalization and the ReLU non-linearity), we have $M_n = (L + \text{Id})^{n-1}$. The eigenvalues of $M_n$ are $(\lambda_i + 1)^{n-1}$, where $\lambda_1, ..., \lambda_d$ are the eigenvalues of $L$. At initialization, the components of $L$ are sampled from a random uniform distribution. To reduce the magnitude of $\lambda_i$, one option is to make the entries of $L$ small. However, to decrease $(\lambda_i + 1)^{n-1}$ sufficiently, $\lambda_i$ must be so small that it introduces floating-point cancellations when the residual block is added back to the shortcut connection. This is why we prefer to adjust $n$, as detailed next.

### 3.3 Progressive training

We adopt a progressive learning schedule in a "warm up" phase: we start the optimization with a single block and add one iteration at every epoch until we reach the required $n$ blocks. This is

| $n_{\mathrm{tr}}$ | Gaussian noise (std=30) | | Gaussian blur ($\sigma$=4) | |
|---|---|---|---|---|
| | POL | ind | POL | ind |
| 1 | 23.3 | **23.3** | 18.6 | 18.6 |
| 4 | 24.4 | 23.2 | 19.2 | 19.2 |
| 8 | 23.9 | 22.3 | 19.0 | **19.3** |
| 12 | 23.9 | 22.5 | **19.7** | 18.8 |
| 16 | 23.9 | 22.5 | 19.0 | 18.1 |
| 18 | **24.2** | – | 19.0 | – |
| 30 | 23.5 | – | 19.0 | – |

Table 1: Denoising: PSNR on Urban-100 Huang et al. (2015). Comparison between Power of layers (POL) and independent (ind) blocks for different maximum number of composition / residual blocks. The best value for each column is in **Bold**. We could not fit more than 16 independent blocks in memory in our experiments. Appendix A reports standard deviations and more results.

possible because the blocks operate in the same embedding space $\mathcal{E}$ and because their weights are shared, so all blocks are still in the same learning schedule. This approach avoids the numerical exponentiation effects at initialization mentioned in the previous paragraph. After the warm-up phase, the gradient descent starts to converge, so these numerical effects become unlikely. In addition, this approach allows the discriminator to improve progressively during the training. For example, in the case of the transformation horse $\rightarrow$ zebra, a slightly whitened horse fools the discriminator at the beginning of the training, but a texture with stronger stripes is required later on.

**Training for modulation.** If the network is trained with a fixed number of compositions, the intermediate states do not correspond to modulations of the transformation that "look right" (see Appendix A). Therefore, in addition to this scheduled number of iterations during the first $n$ epochs of warm-up, we also randomize the number of iterations in subsequent epochs. This forces the generator to also produce acceptable intermediate states, and enables modulating the transform.

**Stopping criterion at inference time.** Each image of domain $\mathcal{A}$ is more or less close to domain $\mathcal{B}$. For example, when denoising, the noise level can vary so the denoising strength should adapt to the input. Similarly for horse$\rightarrow$zebra: a white horse is closer to a zebra than a brown horse. Therefore, at inference time, we can adjust the number of compositions as well. In particular, for each test image, we select $n$ that best deceives the discriminator, thus effectively adapting the processing to the relative distance to the target domain.

## 4 ANALYSIS

In this section we study the impact of the main training and design choices and on the performance of powers of layers. Appendices A and B provide complementary analysis for training- and inference-time choices, respectively. For this preliminary analysis, we focus on denoising tasks for which the performance is easily measurable. We add three types of noise to images: Gaussian noise, Gaussian blur and JPEG artifacts. The noise intensity is quantified by the noise standard deviation, the blur radius and the JPEG quality factor, respectively. We generate transformed images and measure how well our method recovers the initial image. This provides a more reliable accuracy measure than a purely unpaired evaluation, that will be carried out in the experimental section.

Note that, in the literature, these tasks are best addressed by providing (original, noisy) pairs if images. Our objective is to remain in a completely unpaired setting during the training phase. It corresponds to the case where parallel data is not available (like for the restoration of ancient movies), and also better reflects the situation where the noise strength is not known in advance. Therefore, the original image is solely employed to measure the performance.

**Experimental protocol.** To train, we sample 800 domain $\mathcal{A}$ images from the high-resolution Div2K dataset (Agustsson & Timofte, 2017). In the baseline training procedure, the warm-up phase starts from a single block and increases the number of compositions at every epoch, until we reach epoch $n_{\mathrm{tr}}$. Then we keep the number of compositions fixed.

We test on the Urban-100 (Huang et al., 2015) dataset. Unless specified otherwise, we set the number of compositions to $n_{\mathrm{te}} = n_{\mathrm{tr}}$ and measure the Peak Signal to Noise Ratio (PSNR) of our model on the dataset images, degraded with the same intensity as at training time. For the JPEG case we use the Naturalness Image Quality Evaluator metric (NIQE, lower=better) instead, because it is more sensitive to JPEG artifacts (Mittal et al., 2013). NIQE is a perceptual metric that does not take the original image into account (i.e. for JPEG we emphdo use an unpaired metric).

**Block composition or independent successive blocks?** Table 1 compares PoL's composition of the same block versus using independent blocks with distinct weights. In spite of the much larger

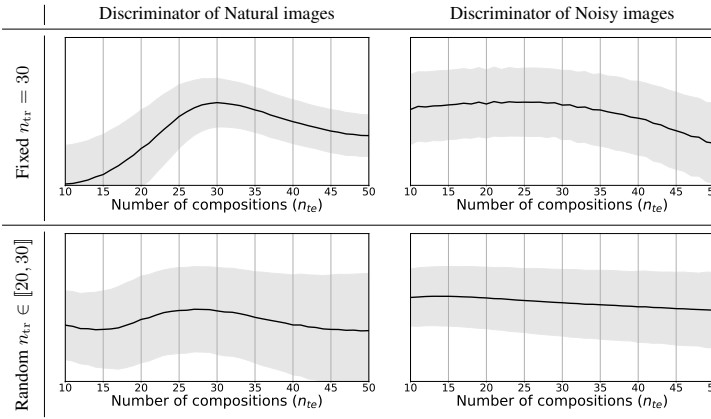

Figure 2: Response of the discriminators as a function of the number of compositions for the transformation Gaussian noise→natural image. We plot the average response and the standard deviation over examples (gray). Higher=the discriminator classifies the image into its target domain. *Top:* training with a fixed number of compositions, $n_{\text{tr}}$=30. *Bottom:* training with randomised $n_{\text{tr}} \in [\![20, 30]\!]$.

capacity offered by independent blocks, the noise reduction operated by Power of layers is stronger. Our interpretation is that the model is easier to train.

**Analysis of the progressive training strategy.** Table 1 also evaluates the impact of the maximum number of compositions $n_{\text{tr}}$. Having several compositions clearly helps. Since we choose the number of compositions $n_{\text{te}}$ at inference time (see next paragraph), it may be relevant to vary $n_{\text{tr}}$ at training time to minimize the discrepancy between the train- and test-time settings. For this, we tried different intervals to randomly draw the maximum number of compositions for each epoch, after the warm-up phase. If $n_{\text{te}}$ is fixed, the optimal choice is $n_{\text{tr}} = n_{\text{te}}$. However, if we use an adaptive $n_{\text{te}}$, the best range is $n_{\text{tr}} \in [\![20, 30]\!]$, and the adaptive case with randomised training gives the best performance for denoising and debluring. Appendix A reports results obtained with different $n_{\text{tr}}$ ranges.

**Stopping criterion.** We consider two cases: either we use a fixed $n_{\text{te}}$, or we use the discriminator to evaluate the transformation quality: it selects the value $n_{\text{te}}$ maximizing the target discriminator error for a given image. Figure 2 shows that setting a fixed $n_{\text{tr}}$ causes the discriminator to select $n_{\text{te}} = n_{\text{tr}}$ as the best iteration at inference time. By selecting the best $n_{\text{te}}$ for each image we obtain on average a PSNR improvement of +1.36dB for a Gaussian noise of standard deviation 30, compared to fixing $n_{\text{te}}$. In Appendix B, we compare it with the best possible stopping criterion: an oracle that selects $n_{\text{te}}$ directly on the PSNR. Our adaptive strategy significantly tightens the gap to this upper bound.

**Comparison with CycleGAN.** We use CycleGAN as a baseline. The differences between CycleGAN and powers of layers are (1) we use a single encoder and decoder trained in advance and common to all tasks; (2) CycleGAN has 9 residual blocks, PoL iterates a single residual block an arbitrary number of times. The inference time of PoL depends on the number of compositions $n_{\text{te}}$ but the number of parameters does not:

|  | encoder + decoder | residual block | discriminators | total |
|---|---|---|---|---|
| PoL | $1\times$ 1.7M | $2 \times K \times$ 1.1M | $2\times$2.7M | 15.9M |
| CycleGAN | $2\times$ 11.4M | | $2\times$2.7M | 28.2M |

Figure 3 compares the performance obtained by the two methods on denoising tasks with varying noise intensities. PoL gives better results than CycleGAN in terms of objective metrics, and overall the images produced by our method look as realistic and/or accurate.

**Training with few images.** Figure 3 also compares our method with CycleGAN when training in a data-starving scenario. Whatever the number of training images, PoL outperforms CycleGAN, but the gap is particularly important with very few images. This is expected for two reasons. Firstly, our approach has fewer parameters to learn than CycleGAN. Secondly, it only requires to learn the transformation because the encoder and decoder are pre-learned as a vanilla auto-encoder, while CycleGAN needs to learn how to encode and decode images.

**Parametrization: remarks.** Beyond the settings inherited from CycleGAN, the main training parameters of Powers of layers are the maximum number of compositions $n_{\text{tr}}$ and the range from which they are randomly sampled. The number of compositions at inference time $n_{\text{te}}$ is also important but the discriminator criterion can be used to set it automatically.

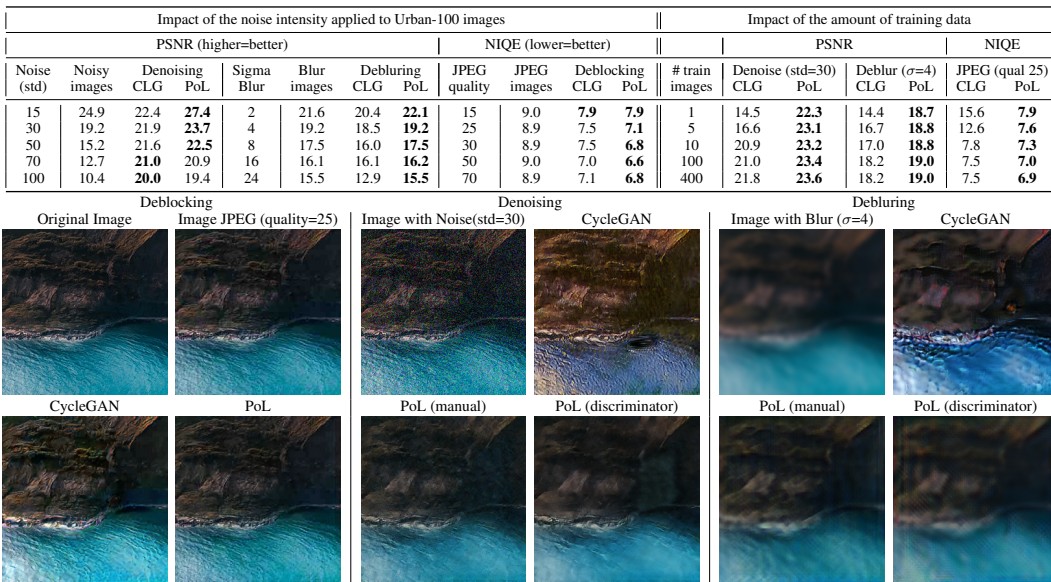

| | | | Impact of the noise intensity applied to Urban-100 images | | | | | | | | | | | | Impact of the amount of training data | | | | | |
|---|---|---|---|---|---|---|---|---|---|---|---|---|---|---|---|---|---|---|---|---|
| | | PSNR (higher=better) | | | | | | | NIQE (lower=better) | | | | | PSNR | | | | | NIQE | |
| Noise (std) | Noisy images | Denoising CLG | PoL | Sigma Blur | Blur images | Debluring CLG | PoL | JPEG quality | JPEG images | Deblocking CLG | PoL | # train images | Denoise (std=30) CLG | PoL | Deblur (σ=4) CLG | PoL | JPEG (qual 25) CLG | PoL |
| 15 | 24.9 | 22.4 | **27.4** | 2 | 21.6 | 20.4 | **22.1** | 15 | 9.0 | **7.9** | 7.9 | 1 | 14.5 | **22.3** | 14.4 | **18.7** | 15.6 | **7.9** |
| 30 | 19.2 | 21.9 | **23.7** | 4 | 19.2 | 18.5 | **19.2** | 25 | 8.9 | 7.5 | **7.1** | 5 | 16.6 | **23.1** | 16.7 | **18.8** | 12.6 | **7.6** |
| 50 | 15.2 | 21.6 | **22.5** | 8 | 17.5 | 16.0 | **17.5** | 30 | 8.9 | 7.5 | **6.8** | 10 | 20.9 | **23.2** | 17.0 | **18.8** | 7.8 | **7.3** |
| 70 | 12.7 | **21.0** | 20.9 | 16 | 16.1 | 16.1 | **16.2** | 50 | 9.0 | 7.0 | **6.6** | 100 | 21.0 | **23.4** | 18.2 | **19.0** | 7.5 | **7.0** |
| 100 | 10.4 | **20.0** | 19.4 | 24 | 15.5 | 12.9 | **15.5** | 70 | 8.9 | 7.1 | **6.8** | 400 | 21.8 | **23.6** | 18.2 | **19.0** | 7.5 | **6.9** |

Figure 3: *Top:* Comparison between Powers of layers (PoL) and CycleGAN (CLG) to denoise images of Urban-100 Huang et al. (2015). We provide standard deviation and additional results in Appendix C. *Bottom:* Visual comparison between our method (manual and discriminator choice) and CycleGAN for deblocking, denoising and debluring.

| Domain | CycleGAN | POL/CycleGAN | NiceGAN | POL/NiceGAN |
|---|---|---|---|---|
| Summer → Winter | 48.8 | **46.1** | **40.5** | 41.9 |
| Summer ← Winter | 48.4 | **44.4** | 39.9 | **39.5** |
| Horse → Zebra | 89.7 | **53.0** | **44.8** | 45.7 |
| Horse ← Zebra | **110.5** | 112.3 | 69.1 | **62.2** |
| Van-Gogh → Picture | 163.4 | **134.4** | **101.4** | 102.8 |
| Van-Gogh ← Picture | **151.4** | 152.7 | 116.5 | **114.9** |
| Cezanne → Picture | **127.4** | 138.8 | **91.1** | 92.2 |
| Cezanne ← Picture | **145.5** | 147.6 | 126.1 | **122.9** |
| Monet → Picture | **60.3** | 70.3 | 65.7 | **63.5** |
| Monet ← Picture | **61.8** | 82.1 | **53.3** | 55.4 |
| Apple → Orange | 88.9 | **83.2** | **112.7** | 114.6 |
| Apple ← Orange | 116.7 | **113.2** | 84.8 | **80.7** |

Figure 4: Left: Frechet Inception Distance (FID) for image-to-image translation tasks (lower is better). We compare CycleGAN and NiceGAN with POL/CycleGAN and POL/NiceGAN, where the respective translation functions are replaced with POL blocks. Right: example results.

## 5 EXPERIMENTS

We now run experiments on two image generation applications. We refer to Section 3.1 for the architecture and training protocol. In Appendix C we also give a comparison with the Fader network for the capacity to modulate a transformation, and more visual examples in Appendix D.

**Unpaired image-to-image translation.** We report results for 6 of the 8 unpaired image-to-image translation tasks introduced in the CycleGAN (Zhu et al., 2017) paper (the two remaining ones lead to the same conclusions) and we used the datasets from their website. We compare the Frechet Inception Distance (FID) of CycleGAN, NiceGAN and our approach applied to this method in Figure 4. The FID measures the similarity between two datasets of images, we use it to compare the target dataset with the transformed dataset. It is a noisy measure for which only large deviations are significant. Yet the results and visualization show that our method has results comparable to those of CycleGAN and NiceGAN, achieved with much fewer parameters.

**High resolution experiments.** PoL is fully convolutional architecture, therefore it is technically possible to apply models trained in a low resolution to high resolution images. However, the results

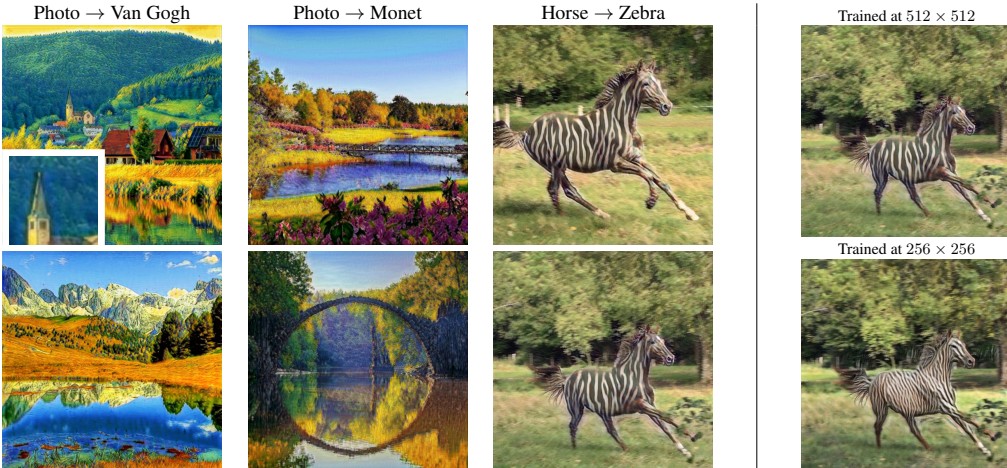

Figure 5: *Left:* Different visual results with high resolution images. See the original images in Appendix D. *Right:* Comparisons of the generations obtained by models trained either with high-resolution or low-resolution images, applied to a high-resolution test image.

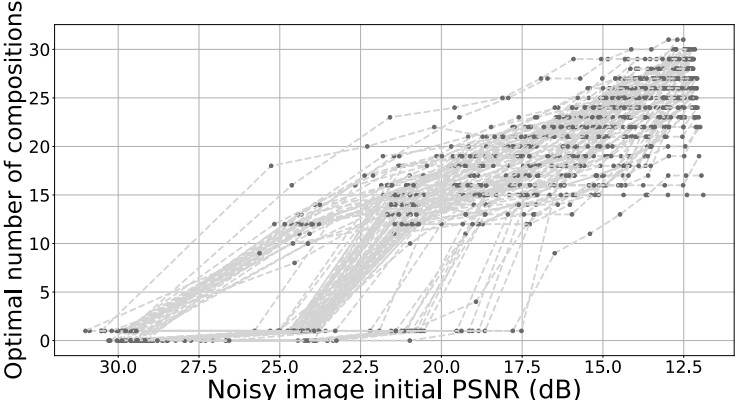

Figure 6: Evolution of the optimal number of composition according to the noise. The maximum number of compositions used for training is 30 and the Gaussian Noise standard deviation is 30.

are not always convincing, as shown in Figure 5 (right) where the model trained on low resolution images does not create stripes at the "right" scale on zebras. To circumvent this problem, CycleGAN trains on patches taken from high resolution images. This works for transformations affecting the whole image (painting↔photo), but this is not applicable in the case where only a part of the image is affected (horse→zebra). In contrast, our proposal can adapt the memory used by changing its number of compositions, so we can apply it to very large images without running out of memory. Figure 5 (left) depicts results obtained with our method trained on high resolution images.

**Adjusting $n_{te}$ at inference time.** Each image is more or less distant from the target domain, so we explore adapting the transformation to each image rather than applying a fixed transformation. For example, depending on the amount of noise, we may want to adjust the strength of the denoising. By modulating $n_{te}$ we can adapt the transformation to each image. Figure 6 shows that the more noisy the input image is, the more we should compose to best denoise with Powers-of-Layers.

In the same way, Figure 7 shows the progressive transformations obtained on different unpaired image-to-image translation tasks. It shows that progressive transformations are realistic for most tasks.

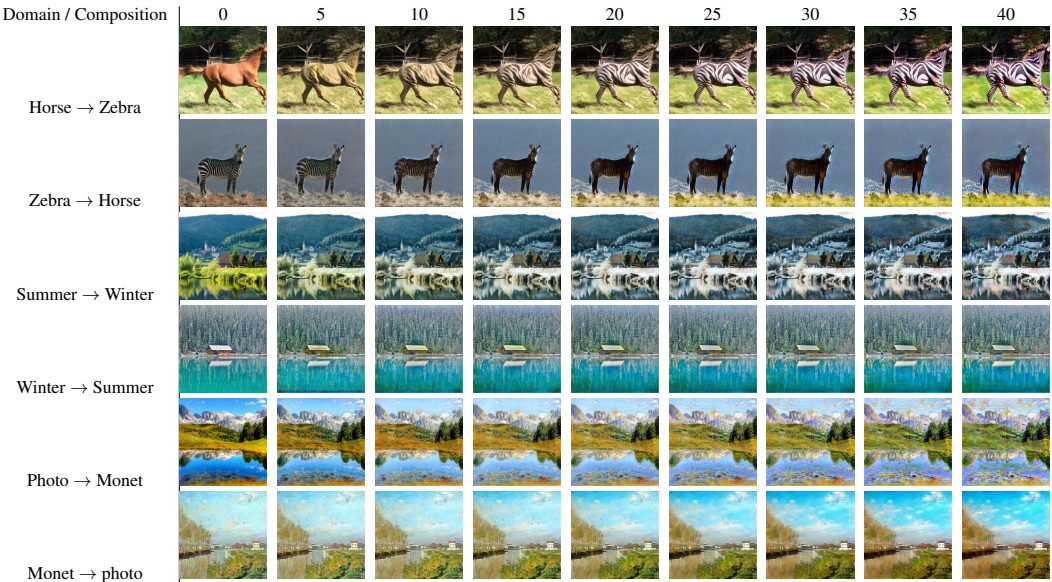

Figure 7: Illustration of the different results obtained along the iterations of the recurrent block, during the transformation of a horse into a zebra. The first image is the original image, then each image corresponds to 5 additional compositions of our method. Since our method was learned with a maximum of 30 compositions the last two images are extrapolations.

**Combining transformation.** The different blocks associated with different transformations operate in the same embedding space for different tasks. Hence we can compose transformations, each being realized by one residual block. We train Transform #1 in the usual way, then freeze its residual block. Transform #2 is trained on the output of #1. Visual results are in Figure 8. The composition in the embedding space gives better results than decoding/encoding to image space mid-way.

## 6 CONCLUSION

Powers of layers iterates a residual block to learns a complex transformation with no direct supervision. On various tasks, It gives similar performance to CycleGAN and NiceGAN with fewer parameters. The flexibility offered by the common embedding space allows the modulation of the transformation strength, or to compose several transformations. While in most examples the discriminator is only used for training, Powers of layers can also exploit it to adjust the transformation to the input image at inference time.

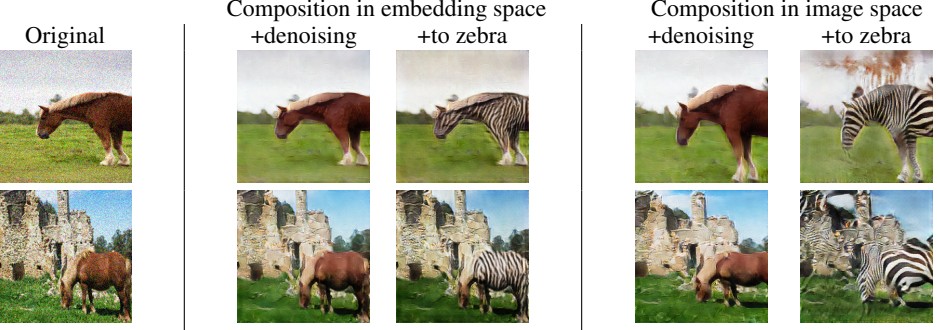

Figure 8: Composition of transformations in the embedding/image space.

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
