# OpenReview forum: "Powers of layers for image-to-image translation"
_ICLR.cc/2021/Conference — Reject_

### Official Review · AnonReviewer3 · 2020-10-15

**Rating:** 3
**Confidence:** 3

**Review:**

The paper proposes a method for unsupervised image translation between unpaired domains of images. The main idea is to develop an iterative transformation module that operates in the embedding space.

Overall I have the following concerns about the paper:

The motivation for this architecture is unclear. The introduction motivates this model with fractals and iterated function spaces, but that seems to have nothing to do with the types of applications shown here. What do IFSs have to do with denoising and pictures of zebras? This iterated refinement strategy seems more similar to iterative refinement/projection algorithms like conjugate gradient and Richardson-Lucy deconvolution, which are relevant to low-level signal processing operations like denoising/deblurring, but not zebra synthesis.  The paper includes as motivation the idea that applying that different levels of transformation can be achieved by choosing different numbers of iterations, but the application of this is shown only for denoising (Table 1).

No comparison is provided to the state-of-the-art in unpaired image translation:
Contrastive Learning for Unpaired Image-to-Image Translation
Taesung Park, Alexei Efros, Richard Zhang, Jun-Yan Zhu
ECCV 2020

Visually, the results are not convincing. Not many results are shown, and most do not look better than those from CycleGAN.  The results may be cherry-picked, since there was no statement as to how these results were chosen.  There is simply not enough visual evidence that the method has evidence over previous work. Additionally, quantitatve comparisons do not give a compelling outcome, but I would put more weight on visual comparisons anyway.

For the task of denoising, it is unclear why one would want to use a general-purpose unpaired translation method; supervised methods ought to be much effective here, and there is an enormous literature of related work that is not cited or compared with here. If one is to use denoising as a motivation application (rather than a toy example), then much more rigor is required.

---

> ### Author Response · Authors · 2020-11-23
> **Response to R3**
>
> 1)  The review states that  “The paper includes as motivation the idea that applying that different levels of transformation can be achieved by choosing different numbers of iterations, but the application of this is shown only for denoising (Table 1).”
> This is not correct. “If the network is trained with a fixed number of compositions, the intermediate states do not correspond to modulations of the transformation that “look right” (see Appendix A)” (paragraph “Training for modulation” page 5) See Figure 8 Appendix A. “In Appendix C we also give a comparison with the Fader network for the capacity to modulate a transformation, and more visual examples in Appendix D.” (Section 5 page 7)  see Appendix D Figure 13 we provide example with 6 different transformations.
>
> 2) See appendix for additional visualization. Images in all our illustrations are not cherry-picked, but overall the objective assessment (PSNR, FID, NIQE) quantitatively show that our method provides some improvement .
>
> 3)  The paper “ Contrastive Learning for Unpaired Image-to-Image Translation” Taesung Park, Alexei Efros, Richard Zhang, Jun-Yan Zhu ECCV 2020 is no more the state of the art in unpaired image to image translation than the paper “Reusing Discriminators for Encoding: Towards Unsupervised Image-to-Image Translation” Chen et al. CVPR 2020 with which we compare our method. Indeed, in Unpaired Image to Image translation, we have 6 tasks in common with  Chen et al. (on the 8 tasks presented in the paper ) and only 1 task in common  with Park et al. (on the 3 tasks presented in the paper). In addition, as you can see in Table 1 in Park et al. they do not provide a comparison with Chen et al. and are evaluated in different ways so make it impossible to judge which of the two methods is the best (Compare Table 1 in Chen et al. and Table 1 in Park et al.).
> Given all these elements and having more tasks in common with Chen et al. , we found it more appropriate to use this method as our baseline to compare with the state of the art. We have also proposed a version of PoL applied to CycleGAN's architecture (Unpaired Image-to-Image Translation using Cycle-Consistent Adversarial Networks, Zhu et al.), as it is a very popular architecture.
>
> 4) The objective of our analysis section is not to have state of the art results in denoising and deblurring, which one can not expect from a super generic method that can handle multiples tasks with a single architecture (therefore more compact and more easily extendable, and less prone to overfit to a specific amount of noise or blur). In contrast, this analysis shows an experimental context that is easier to evaluate than the experimental part in order to better understand the impact of the different components of the method.

---

### Official Review · AnonReviewer2 · 2020-10-25
**A good point to reduce the number of weights but the computation time is questionable**

**Rating:** 5
**Confidence:** 4

**Review:**

1. Summary. The submitted paper proposes to use a recurrent residual block for the task of unpaired img2img translation. A number of strategies of how many times to apply this block are suggested.

1. Decision. I do really like the direction of weight sharing in img2img models and this work is a pretty nice case. This approach helps to decrease the number of weights, and, if done properly, does not harm quality. The results in deblurring/denoising look rather interesting.

Cons.

However, the downside of the presented recurrent block is the increased computation time at the inference step, as far as I can judge. This is especially crucial when the discriminator is involved as the stopping criterion. Could you provide a comparison of the inference speed (FPS or FLOPS or any other measure) between CycleGAN/NiceGAN and PoL?

Second, I believe this approach could be also put into the context of the adaptive computation time research field [1,2,3,4]. This may help to determine the number of layers to apply.

Third, the proposed block may be straightforwardly generalized to multi-domain img2img translation and showcased on more interesting and recent datasets against stronger baselines like MUNIT [5], FUNIT[6], etc. This could make img2img part of your experiments more solid, I suppose.

All-in-all, to my mind there is great room for improvement for your submission to demonstrate the real power of PoL. Therefore, now I tend to rate the submission a bit below the threshold.

[1] https://openreview.net/forum?id=r1W1OxAF
[2] https://openreview.net/forum?id=SkZq3vyDf
[3] https://openaccess.thecvf.com/content_cvpr_2017/html/Figurnov_Spatially_Adaptive_Computation_CVPR_2017_paper.html
[4] https://openreview.net/forum?id=HyzdRiR9Y7
[5] https://link.springer.com/chapter/10.1007/978-3-030-01219-9_11
[6] https://openaccess.thecvf.com/content_ICCV_2019/html/Liu_Few-Shot_Unsupervised_Image-to-Image_Translation_ICCV_2019_paper.html

---

> ### Author Response · Authors · 2020-11-23
> **Response to R2**
>
> 1 ) The reviewer is correct that, if we use the stopping criterion at each iteration, then the overall computation time would be higher. In practice, as shown by Table 3 and Figure 10 in Appendix A, the interesting range for iterations is relatively limited and we can restrict to number of times we use the discriminator
> It is possible to set the maximum number of compositions in POL to the number of residual blocks present in CycleGAN or NiceGAN. So in this case we have the same inference time but fewer parameters. In this case as we have fewer parameters we can be faster than the baseline by using larger batches. We do this with high resolution images.
>
> 2 & 3) Thanks for the references and the suggestion to improve our submission

---

### Official Review · AnonReviewer4 · 2020-10-27
**Insufficient comparison and experiment settings**

**Rating:** 5
**Confidence:** 4

**Review:**

This paper proposes an unpaired image-to-image translation method which applies a pre-trained auto-encoder and a latent feature transformer (single block) to perform iterative image transformation. A progressive training and warm-up strategy is used to settle the numerical exponentiation effects caused by powers of layers. In the testing phase, the discriminator is also used to adjust the inference time.

Pros:
1) Compared with the vanilla CycleGAN, the proposed PoL has significantly fewer parameters and similar performance.
2) The flexibility offered by the common embedding space allows the modulation of the transformation strength, or to compose several transformations.
3) The discriminator is used to adjust the inference time and find the optimal number of iterations in the testing phase.

Cons:
1) CycleGAN is the only competitor in most comparative experiments, which is not sufficient. Besides, CycleGAN is not a good competitor for image restoration tasks (debluring, denoising, etc.), so the potential of the proposed PoL is questionable. Additional comparison results generated by other general image restoration methods [1, 2] should be reported.
2) Is CycleGAN also pre-trained on the same dataset as PoL for fair comparison?
3) Although progressive training contributes to more natural intermediate outputs, the final output is not satisfactory, for example the unnatural patterns on zebras in Fig. 4.
4) More effective transformation modulation is a major advantage of the proposed PoL, but the provided experiments did not demonstrate this well. I think it would be more appropriate to put the results obtained along the iterations of the recurrent block in the main text instead of in the appendix.
5) Why the proposed embedding transformer “is similar to the feed-forward network used in transformers [3]”? It seems that it is just a simple residual convolution module with expansion factor K to adjust the model’s capacity, which is not related to transformer or self-attention.
6) What are the significant advantages of PoL compared with traditional RNNs (LSTM, GRU)? Is it possible to directly replace PoL with RNNs to achieve close results?

[1] Neural Sparse Representation for Image Restoration. NeurIPS, 2020.
[2] Learning Invariant Representation for Unsupervised Image Restoration. CVPR, 2020.
[3] Attention is all you need. NeurIPS, 2017.

---

> ### Author Response · Authors · 2020-11-23
> **Response to R4**
>
> We thank the reviewer for her/his feedback.
>
> 1) CycleGAN is the only competitor for these tasks because we do not target a single pretext tasks (denoising deblurring), as we mostly focus on comparing architectures designed for Unpaired Image translation  (e.g. CycleGAN and PoL), as discussed in the introduction. This allows the most fair possible comparison.
> Requiring our generic method (which has not seen any paired data and can address multiple tasks) to be better than specific architectures specifically optimized for a single task, with paired data reflecting a specific level of noise of blur (while our method can accommodate a range as shown in our paper), does not sound reasonable to us.
>
> 2) We explain this in Section 3.1
>
> 3) Nevertheless, the result is visually more accurate than the CycleGAN result. The FID measurements indicate that we have equivalent results with fewer parameters.
>
> 4)  We are pleased to add these results in the updated  9 pages submission.
>
> 5) The principle of using an expansion factor between two blocks is what is used in the MLP of the feedforward part of the transformer.
>
> 6) There are several differences and hurdles that would need to be addressed with a traditional RNN, which we actually address in our paper. (i) RNN are not spatial, so one would need to adapt them to process properly the 2d+depth feature map. (ii) one must treat properly how many times you apply the residual blocks, and how you train it.
> Overall, the gating in modern RNNs aims at enlarging the context and avoids catastrophic forgetting effects, which is different from the unpaired domain transfer that we consider. While it may be possible to apply RNN to unpaired domain adaptation, it is not straightforward.  For instance, to learn translation between two languages without having parallel sentences, recent works (See works by Lample, Conneau et al.[1] for instance) don’t use a direct application of RNNs. Therefore, this would require some substantial work to make it work, which we achieved in our paper with our simple PoL architecture.
>
> [1]  Guillaume Lample, Alexis Conneau, Marc'Aurelio Ranzato, Ludovic Denoyer, Hervé Jégou ,Word translation without parallel data, ICLR 2018

---

### Official Review · AnonReviewer1 · 2020-11-01
**reviews for Powers of layers for image-to-image translation**

**Rating:** 5
**Confidence:** 4

**Review:**

This paper presents an approach for image to image translation by introducing extra layers into the generator, which can be trained in an unsupervised way. The paper is generally easy to follow. I have the following concerns: the novelty is quite marginal since the backbone network and the training process are well developed before and the technique of employing more layers to the generator seems like simply extending the network. Please consider to use different notations in Section 3 to denote image from different domains. Also, please use the same subscript for G_{AB} and G_{\mathcal{BA}}.

---

> ### Author Response · Authors · 2020-11-23
> **Response to R1**
>
> We do not use extra layers for the generator, quite the opposite: we use only one residual block with PoL versus 9 residual blocks for the baseline.
> In our PoL framework, we train once the encoder and the decoder with a reconstruction task; then they are freezed, and we learn the transformations with only 1 residual block.

---

### Decision · Program_Chairs · 2021-01-07
**Final Decision**

**Decision:**

Reject

**Comment:**

All the reviewers shared the concerns about the novelty and the quality of the results. Comparisons with some SOTA results are missing, and the inclusion of deblurring/denosing tasks is not convincing. The authors carefully addressed these issues in the rebuttal but the reviewers didn’t change their mind afterwards. After carefully examining the results in the paper, the AC agrees with the reviewers that the improvement on image quality, if any, seems to be too small to warrant a publication.